# Automatic Counterfeit Currency Detection Using a Novel Snapshot Hyperspectral Imaging Algorithm

**DOI:** 10.3390/s23042026

**Published:** 2023-02-10

**Authors:** Arvind Mukundan, Yu-Ming Tsao, Wen-Min Cheng, Fen-Chi Lin, Hsiang-Chen Wang

**Affiliations:** 1Department of Mechanical Engineering, Advanced Institute of Manufacturing with High Tech Innovations (AIM-HI) and Center for Innovative Research on Aging Society (CIRAS), National Chung Cheng University, 168, University Rd., Min Hsiung, Chia Yi 62102, Taiwan; 2College of Management, National Sun Yat-sen University, 70 Lienhai Rd., Kaohsiung 80424, Taiwan; 3Medical Devices R&D Service Development, Metal Industries Research & Development Centre, 1001 Kaonan Highway, Nanzi District, Kaohsiung 80284, Taiwan; 4Ophthalmology, Kaohsiung Armed Forces General Hospital, 2, Zhongzheng 1st.Rd., Lingya District, Kaohsiung 80284, Taiwan; 5Director of Technology Development, Hitspectra Intelligent Technology Co., Ltd., 4F, No.2, Fuxing 4th Rd., Qianzhen District, Kaohsiung 80661, Taiwan

**Keywords:** hyperspectral imaging, fake currency, mean gray value, Raspberry Pi, region of interest

## Abstract

In this study, a snapshot-based hyperspectral imaging (HSI) algorithm that converts RGB images to HSI images is designed using the Raspberry Pi environment. A Windows-based Python application is also developed to control the Raspberry Pi camera and processor. The mean gray values (MGVs) of two distinct regions of interest (ROIs) are selected from three samples of 100 NTD Taiwanese currency notes and compared with three samples of counterfeit 100 NTD notes. Results suggest that the currency notes can be easily differentiated on the basis of MGV values within shorter wavelengths, between 400 nm and 500 nm. However, the MGV values are similar in longer wavelengths. Moreover, if an ROI has a security feature, then the classification method is considerably more efficient. The key features of the module include portability, lower cost, a lack of moving parts, and no processing of images required.

## 1. Introduction

Counterfeit currency notes are being rapidly circulated in every economy in recent years [1,2]. In the last decade, more than 3.53 lakh cases were reported due to banknote counterfeiting [3]. In the USA alone, a loss of more than 20 billion USD has been projected due to counterfeit checks [4]. With recent developments in counterfeiting technology, the human eye has lost the ability to differentiate between original and counterfeit banknotes [5].

To prevent counterfeiting of banknotes, every monetary authority has its own security features. For example, the national central banks of the euro system use micro-perforations [6]. In the hologram patch of the euro, the “€” symbol is formed by micro-perforations, or micro-holes. The Reserve Bank of India (RBI) uses the Omron anti-photocopying feature, which appears in yellow circles on either side of the text “Reserve Bank of India” [7]. In 1996, the USA added a few security features, including a 3D security ribbon and a color-shifting bell in the ink [8]. When the currency is tilted, the bell and “100” will change their positions.

Even with many anti-counterfeiting techniques, the amount of counterfeit currencies has still increased in recent years [5,9] due to technological evolutions, such as color printing, duplication, and scanning [10]. Current commercially available techniques use ultraviolet (UV) light to detect ink marks that are invisible to the human eyes [5,11]. However, this process is slow and taxing. In contrast with other techniques, it requires a human operator who knows exactly where a security feature is located on a banknote to verify authenticity, and thus is prone to human error. 

Even though hyperspectral imaging (HSI) systems have previously used in many other counterfeit detection protocols, one optical method that has not been widely studied for counterfeit currency detection is HSI [12,13,14,15]. In one study, a non-destructive analysis and authentication of written and printed documents was performed with a VIS-HSI imaging technique [16]. In another study, near-infrared hyperspectral imaging (NIR-HSI) was developed to detect fraudulent documents [17]. HSI acquires the spectrum for each pixel in an image [18,19,20,21]. It has been used in many applications, such as cancer detection [22,23,24,25], air pollution monitoring [26,27], nanostructure identification [28,29,30,31], aerospace [32,33,34], food quality maintenance [35], verification [36,37,38], military [39], remote sensing [40,41,42], and agriculture [43]. 

Automatic recognition is a prerequisite for any counterfeit-currency detection method. In the past, many counterfeit-currency-note detection techniques have been developed, including UV light detection [44], image processing [11,45,46], deep learning [47,48], bit-plane slicing [49], and artificial neural networks [50]. However, most of these techniques have remained within laboratories because they require complex preprocessing. Moreover, HSI is not currently employed in counterfeit currency detection. Therefore, in this study an automatic counterfeit-currency detection application based on HSI is designed and developed. 

In the current study, a portable low-cost HSI is designed to capture an image of a Republic of China (ROC) banknote and classify it by extracting three regions of interest (ROIs) from the banknote and measuring the mean gray value (MGV). The Section 2 explains the security features of a Taiwanese banknote, the module designed in this study, and the HSI algorithm built to convert an RGB image into a spectral image. The Section 3 describes the extracted ROIs and compares the MGVs of counterfeit and original banknotes. The Section 4 discusses the advantages, limitations, and lessons learned from this study.

## 2. Materials and Methods

### 2.1. Security Features of a Taiwanese Banknote

The national currency of the ROC (also known as Taiwan), called the new Taiwan dollar (NTD), has eight security features, including intaglio printing, optically variable ink (OVI), watermarks, and latent texts, as shown in Figure 1 [51,52,53]. The left side of the Taiwanese 100 NTD banknote has watermarks of orchids and the Arabic numeral of the denomination [54]. In addition, when the note is tilted at an angle of 15°, the Arabic numerical value of the denomination appears to have two different shades. OVI is also used on the bottom left corner of the note, wherein the Arabic numerical value of the denomination appears in two distinct colors (green and purple) when viewed from different angles. Apart from these features, intaglio printing is also added at five separate locations. However, all the aforementioned security features are implemented on the same side of the note.

### 2.2. Experimental Setup

One of the challenges in designing a machine that can detect counterfeit banknotes by using HSI is that it must be portable and low-cost. Most hyperspectral and multispectral machines built thus far are fixed and expensive because they require components, such as a spectrometer, an optical head, and multispectral or hyperspectral lighting systems [55,56,57,58,59]. The novelty of the current study is all the expensive and heavy components are replaced with low-cost and portable units by developing an HSI algorithm that can convert an RGB image, captured by a digital camera, into a hyperspectral image. 

In any optical detection and classification technique, lighting conditions must remain constant, specifically when measuring MGV. Therefore, the second challenge is to design a module such that lighting conditions remain constant while reducing all surrounding light. In such a case, the module must have its own lighting and processing units. 

The schematic of the module designed to capture the image is presented in Figure 2. The whole module is comprised of only six components. Raspberry Pi 4 Model B is used as the microprocessor. Raspberry Pi camera version 2 is used to capture an image of the currency. The camera is connected to Raspberry Pi through a Raspberry Pi mobile industry processor interface, camera serial interface, and camera pinout (see Appendix A for all the components used in this study). An Adafruit thin-film transistor (TFT) display, with 320 × 240 16-bit color pixels in a touch screen measuring 2.8 inches, is added to the module to control Raspberry Pi. In addition, a 3000 K chip on-board (COB) light-emitting diode (LED) strip is fixed along the inside border of the module to provide light for the module. The COB LED strip has a uniform spectral response (no cyan dip) across the blue, green, and red color spectrums. However, this lighting system is not even, but is rather a pointed light source. Therefore, a white opal profile diffuser is used to distribute light evenly. The COB LED strip is also connected to an LED dimmer switch to adjust brightness. The final module is shown in Figure 3.

### 2.3. Visible Snapshot-Based RGB to HSI Conversion Algorithm

The goal of this research is to develop a snapshot-based VIS-HSI imaging algorithm that can convert an RGB image, captured by a point-and-shoot-based image capturing system, into a VIS spectral image. In order to attain this goal, it is necessary to ascertain the connection that exists between the colors included in an RGB image and a spectrometer. A Macbeth chart with 24 basic colors, including red, green, blue, cyan, magenta, yellow, medium light human skin, medium dark human skin, blue sky, foliage, blue flower, bluish green, orange, purplish blue, moderate red, purple, yellow green, orange yellow, white, black, and four different shades of gray, is used as the target colors in order to obtain this relationship. For the purposes of this investigation, the Macbeth chart was selected because it contains the colors that are found in natural settings the most frequently and in the greatest abundance. Due to the fact that a significant amount of research has been conducted using this camera in recent years for the purpose of target color calibration, the Raspberry Pi camera was chosen for the calibration process. The Raspberry Pi camera settings are modified such that any image that is taken will be saved in the sRGB (0–255) color space using the 8-bit JPG file format. To make things easier to understand, the values are first turned into a more compact function ranging from 0 to 1, and then, using the gamma function, they are translated into a linearized RGB value. After that, the values are converted into the color space defined by the CIE 1931 *XYZ* standard using the conversion matrix (*M*). On rare occasions, the camera may be subject to a variety of faults, including color shift, nonlinear response, dark current, erroneous color separation of the filter, and other similar issues (see Appendix A for all the conversion formulas used in this study). The individual conversion formulas to convert the 24-color patch image and 24-color patch reflectance spectrum data to *XYZ* color space on the Raspberry Pi camera side are shown in Equations (1)–(4).
(1)$$\left[\begin{array}{c} X\\ Y\\ Z \end{array}\right]=\left[M_{A}\right]\left[T\right]\left[\begin{array}{c} f\left(R_{sRGB}\right)\\ f\left(G_{sRGB}\right)\\ f\left(B_{sRGB}\right) \end{array}\right]\times 100\;,\;0\leq \begin{array}{c} R_{sRGB}\\ \begin{array}{c} G_{sRGB}\\ B_{sRGB} \end{array} \end{array}\;\leq 1$$
(2)$$\left[T\right]=\left[\begin{array}{c} 0.4104\;0.3576\;0.1805\\ 0.2126\;0.7152\;0.0722\\ 0.0193\;0.1192\;0.9505 \end{array}\right]$$
(3)$$f\left(n\right)=\left\{\begin{array}{c} {\left(\frac{n+0.055}{1.055}\right)}^{2.4},\;n>0.04045\\ \left(\frac{n}{12.92}\right),\;otherwise \end{array}\right.$$
(4)MA=XSWXCW      0       0        0     YSWYCW      0         0         0      ZSWZCW

On the spectrometer side, to convert the reflection spectral data to *XYZ* color gamut space, Equations (5)–(8) are used.
(5)$$X=k\int_{\text{400nm}}^{\text{700nm}}S\left(\lambda \right)R\left(\lambda \right)\bar{x}\left(\lambda \right)d\lambda$$
(6)$$Y=k\int_{\text{400nm}}^{\text{700nm}}S\left(\lambda \right)R\left(\lambda \right)\bar{y}\left(\lambda \right)d\lambda$$
(7)$$Z=k\int_{\text{400nm}}^{\text{700nm}}S\left(\lambda \right)R\left(\lambda \right)\bar{z}\left(\lambda \right)d\lambda$$
(8)$$k=100/\int_{\text{400nm}}^{\text{700nm}}S\left(\lambda \right)\bar{y}\left(\lambda \right)d\lambda$$

The nonlinear response of the camera can be corrected by a third-order equation, and the nonlinear response correction variable is defined as *V_Non-linear_*, as shown in Equation (9).
(9)$$V_{Non\text{-}linear}={\left[X^{3}\;Y^{3}\;Z^{3}\;X^{2}\;Y^{2}\;Y^{2}\;X\;Y\;Z\;1\right]}^{T}$$

In the dark current part of the camera, the dark current is usually a fixed value and does not change with the amount of incoming light, so a constant is given as the contribution of the dark current, and the dark current correction variable is defined as *V_Dark_*, as shown in Equation (10).
(10)$$V_{Dark}=\left[a\right]$$

Finally, *V_Color_* is used as the base, as shown in Equation (11), and is multiplied by the nonlinear response correction of *V_Non-linear_*; the result is standardized within the third order to avoid excessive correction, and finally, *V_Dark_* is added to obtain the variable matrix *V* as shown in Equation (12).
(11)$$V_{Color}={\left[XYZ\;XY\;XZ\;YZ\;X\;Y\;Z\right]}^{T}$$
(12)$$V={\left[X^{3}\;Y^{3}\;Z^{3}\;X^{2}Y\;X^{2}Z\;Y^{2}Z\;XY^{2}\;XZ^{2}\;YZ^{2}\;XYZ\;X^{2}\;Y^{2}\;Y^{2}\;XY\;XZ\;YZ\;X\;Y\;Z\;a\right]}^{T}$$

As stated in Equations (13) and (14), these mistakes can be rectified by making use of a variable matrix, denoted by the letter *V*. This allows one to retrieve the *X*, *Y*, and *Z* values that have been corrected (*XYZ_Correct_*).
(13)$$\left[C\right]=\left[XYZ_{Spectrum}\right]\times pinv\left(\left[V\right]\right)\;$$
(14)$$\left[XYZ_{Correct}\right]=\left[C\right]\times \left[V\right]\;$$

The same Macbeth chart, under a controlled lighting environment, is also supplied to an Ocean Optics QE65000 spectrometer to find the reflectance spectra of the 24 colors. The brightness ratio (k) is obtained from the standardized value of brightness directly from the *Y* value of the *XYZ* color spectrum (*XYZ_Spectrum_*). In addition, *XYZ_Spectrum_* is converted into the CIE 1931 *XYZ* color space. The construction process of this algorithm is illustrated in Figure 4.

The root-mean-square error (RMSE) of all 24 colors, compared with *XYZ_Spectrum_* and *XYZ_Correct_*, is only 0.19, which is such a small value that it is essentially negligible. In addition to multiple regression analyses, the first six principal components are computed by performing a principal component analysis (PCA) on the reflectance spectrum (*R_Spectrum_*) data acquired from the spectrometer. In order to establish the nature of the connection that exists between *XYZ_Correct_* and *R_Spectrum_*, the dimensions of *R_Spectrum_* were reduced. Following completion of the study, the first six principle components were determined to be the primary components, which account for 99.64% of the variation in the data. In the process of conducting a regression analysis on *XYZ_Correct_*, the variable *V_Color_* was selected as the dependent variable because it contains all of the feasible combinations of *XYZ* values. Utilizing Equation (15), we were able to extract the analog spectrum, denoted by *S_Spectrum_*, of the 24-color block, which we then contrasted with *R_Spectrum_*. Converting *XYZ_Correct_* and *XYZ_Spectrum_* from the *XYZ* color space to the CIELAB color space is required prior to making use of CIE DE2000 for the calculation of color difference. The following formula should be used for the conversion:(15)$$\begin{array}{c} L^{*}=116f\left(\frac{Y}{Y_{n}}\right)-16\\ \begin{array}{c} a^{*}=500\left[f(\frac{X}{X_{n}})-f(\frac{Y}{Y_{n}})\right]\\ b^{*}=200\left[f(\frac{Y}{Y_{n}})-f(\frac{Z}{Z_{n}})\right] \end{array} \end{array}$$
(16)$$f\left(n\right)=\left\{\begin{array}{c} \;\;n^{\frac{1}{3}},\;\;\;n>0.008856\\ 7.787n+0.137931,\;otherwise \end{array}\right.$$

The RMSE value of the 24 colors in the Macbeth chart is calculated individually, and the average deviation is 0.75. This value indicates that the reproduced reflectance spectrum is indistinguishable, and the colors are replicated accurately.
(17)$${\left[SSpectrum\right]}_{380–\text{700nm}}=\left[EV\right]\left[M\right]\left[V_{Color}\right]$$

The aforementioned method can be used to accurately reproduce a hyperspectral image from an RGB image captured by a Raspberry Pi camera, eliminating the need for expensive and heavy components, and making the module low-cost and portable. 

### 2.4. Classification Method

In this study, to classify the currency notes, first the Raspberry Pi module is used with the help of the Windows-based Python application to capture the currency note from the Raspberry Pi camera. A ROI is cropped out from the currency notes. Secondly, the wavelength at which the currency should be analysed is selected. Then, the MGV of the ROI at that specific wavelength is measured. Since MGV is an average intensity of all the pixels in the ROI, the orientation or alignment of the ROI does not make a difference in the output value. Based on the MGV value, the currency note is classified as either counterfiet or original.

## 3. Results

In this study, two ROIs are selected for further analysis. These ROIs are cropped from the original image by using the template matching function. The first ROI is a symbol of a cherry blossom. It has a surface area of 1 cm^2^. The second ROI is the number “1” found on the left bottom corner, as shown in Figure 5. ROI 2 has a surface area of 0.336 cm^2^. ROI 1 does not have any specific security feature, while ROI 2 is printed using OVI. Therefore, the sensitivity of the method developed in this study can be compared with the security features of the banknote (see Appendix A for the entropy measurement of ROI 1). 

Since this study represents a pilot study, three counterfeit and three original 100 NTD banknotes are selected for analysis. For both ROIs, the MGV is found in the visible band between 400 nm and 500 nm. The MGV is the average measure of brightness of every pixel in an image; it has been used in various image classification and measurement methods in the past [60,61,62,63,64]. Figure 6 shows the MGV of ROI 1, while Figure 7 shows the MGV of ROI 2 (see Appendix A for the MGV mean of the duplicate and original samples).

Notably, even though the VIS-HSI algorithm can calculate MGVs up to 700 nm, all the original and counterfeit banknotes have similar MGVs after 500 nm (see Appendix A for MGVs in the range of 400–700 nm). Hence, we can conclude that the most suitable wavelength region for this application is between 400 nm and 500 nm. The other reason is that the UV wavelength is the most sensitive region for detecting counterfeit banknotes. Given that 400–500 nm corresponds to the violet and blue spectra, these bands are efficient. The average RMSE in this band is 8.008 in ROI 1 and 14.079 in ROI 2. Therefore, we can conclude that ROI 2 exhibits better classification sensitivity because ROI 2 has security features, such as OVI and intaglio printing, whereas ROI 1 does not have any specific security feature. 

The 90% confidence interval (CI) is calculated around the average MGV of the original and counterfeit banknotes in ROI 1. The original and counterfeit banknotes are classified into two different classes. The 90% CI around the average of each class represents the range in which the MGV value of the banknotes belonging to that specific class will fall in that specific wavelength. Notably, in ROI 2, 95% CI is calculated and the original and counterfeit banknotes are classified into two different classes within this 5%.

To capture the image of a banknote through the designed module, the Raspberry Pi web-camera interface is installed on the Raspberry Pi operating system. In this study, a Python-based Windows application is also developed to capture and analyze images. This application can directly control the Raspberry Pi camera by using the “Start” and “Stop” buttons to start and stop the live feed, respectively. The “Capture” button is used to capture an image. The narrowband wavelength values in which the banknote must be analyzed, and the gain value specified for each wavelength, are the input. Once the “Analyze” button is clicked, the template-matching function crops the selected ROI and converts it into the specified wavelength. The snapshot-based VIS-HSI algorithm developed in this study is applied to the image, and the MGV of the image is measured. Finally, the banknote is classified as either original or counterfeit based on the CI value, and the results are displayed at the bottom of the application, as shown in Figure 8.

## 4. Conclusions

As a point of reference for identifying and categorizing counterfeit banknotes, this study made use of three authentic polymer 100 NTD banknotes, as well as three counterfeit versions of the same note. It was decided to design and 3D print a module that would be capable of housing a Raspberry Pi 4 Model B, a Raspberry Pi camera, a 2.8-inch TFT touch screen, and an LED strip. It was possible to develop a snapshot-based VIS-HSI algorithm that, when applied to an RGB image, could transform it into a hyperspectral image. This image could be taken by the Raspberry Pi camera housed within the module. The mean gray value (MGV) was determined after choosing two ROIs from the banknotes. The 90% confidence interval (CI) was also calculated around the MGVs for both classes, and the samples were categorized based on the range provided by this CI. In addition, a Python-based Windows application was developed for the purpose of analyzing the sample and capturing the image of a hologram. This application could manage the interface of the web camera used by the Raspberry Pi computer. The findings indicated that the MGV values of the various classes were within the range of the 90% confidence interval (CI). The important novelty of this study is the usage of HSI to detect counterfeit currency by converting an RGB image to an HSI image, without the usage of any moving components or an expensive spectral imager. This study has the potential to be expanded further by taking into consideration additional polymer banknotes as well as different ROIs. In addition, using this research to collect reflectance spectra allows for the construction of a spectral library.

## Figures and Tables

**Figure 1 sensors-23-02026-f001:**
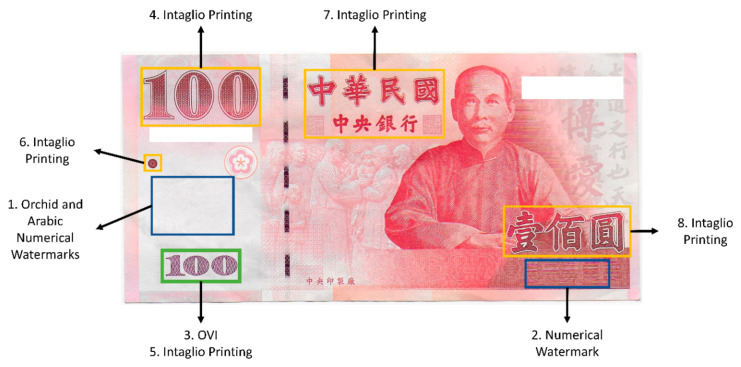
Security features of the 100 NTD banknote.

**Figure 2 sensors-23-02026-f002:**
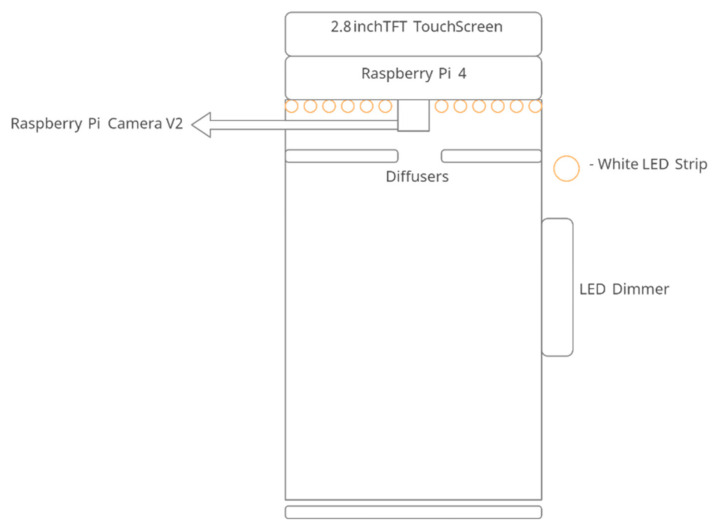
Schematic of the HSI system.

**Figure 3 sensors-23-02026-f003:**
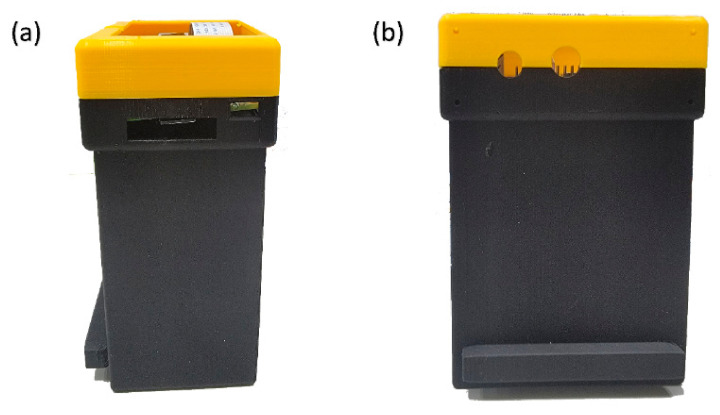
(**a**) Front view and (**b**) side view of the 3D printed design.

**Figure 4 sensors-23-02026-f004:**
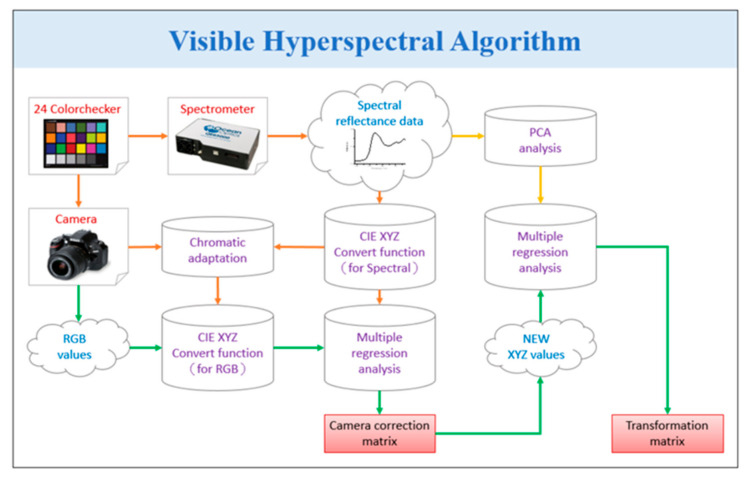
Snapshot-based VIS-HSI algorithm.

**Figure 5 sensors-23-02026-f005:**
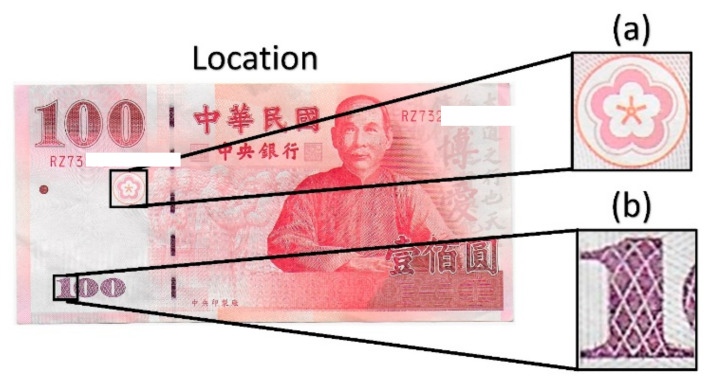
ROI selection: (**a**) cherry blossom symbol and (**b**) Arabic numeral “1”.

**Figure 6 sensors-23-02026-f006:**
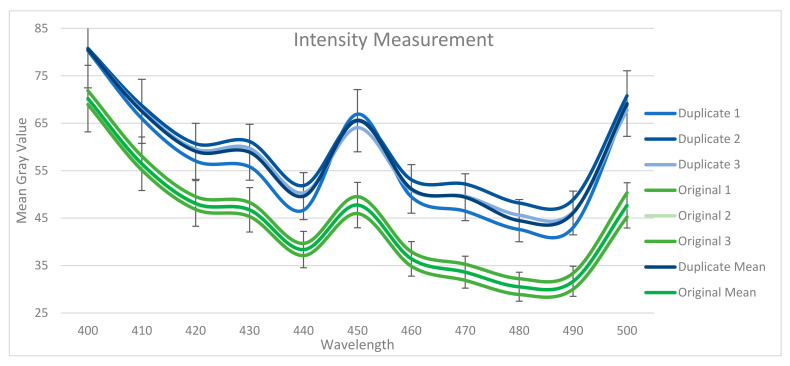
MGVs of the duplicate and original samples in ROI 1.

**Figure 7 sensors-23-02026-f007:**
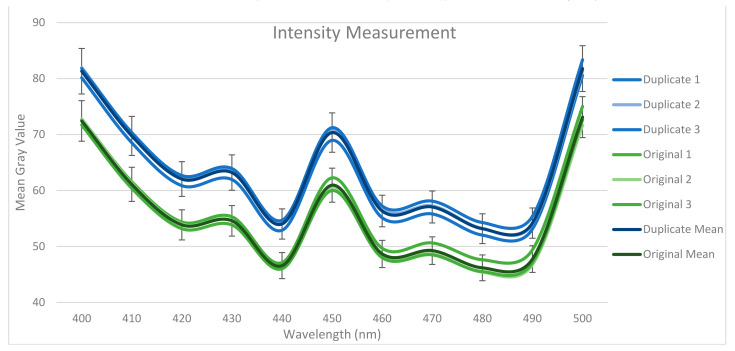
MGVs of the duplicate and original samples in ROI 2.

**Figure 8 sensors-23-02026-f008:**
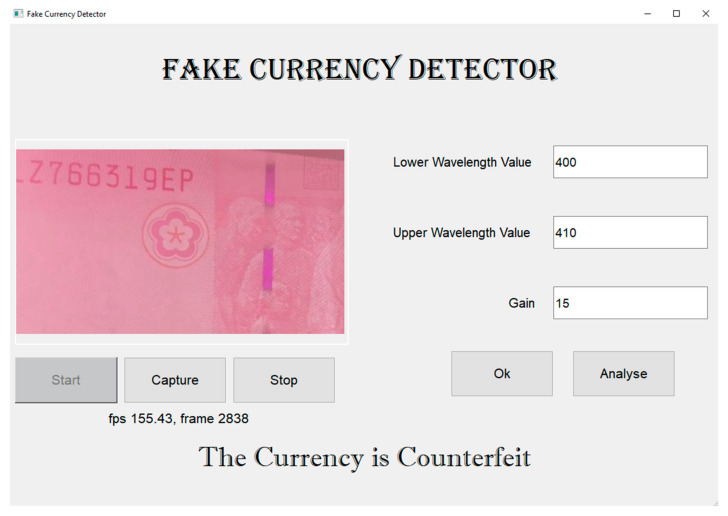
Python-based Windows application for detecting counterfeit currency.

## Data Availability

Not applicable.

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
