# Peer review of "Automatic Counterfeit Currency Detection Using a Novel Snapshot Hyperspectral Imaging Algorithm"

_sensors, 2023, doi:10.3390/s23042026_

Round 1
Reviewer 1 Report
1. Raspberry Pi 4 in Figure 2 seems to be inconsistent with Raspberry Pi 3 Model B written in Line 100, please check it.
2. What do Xn, Yn, and Zn represent in Formula 3? Are they a specific parameter? Please explain if possible.
3. In Fig. 6, the color of Duplicate 1 is similar to Original 2, it is suggested to change it for convenience.
4. The legend in Figure 7 misses the original mean.
5. In Line 208, from Figure 6 and Figure 7, the duplicate sample that is outside the 95% confidence interval seems to be in ROI1 rather than ROI2, please check it.
6. Hyperspectral image processing has been involved in both military and civilization, to better motivate this paper, the authors are suggested to discuss and review more recently published methods, such as:
(1) A Hyperspectral Anomaly Detection Method Based on Low-Rank and Sparse Decomposition with Density Peak Guided Collaborative Representation. IEEE Transactions on Geoscience and Remote Sensing, 2022.
(2) Interpretable Hyperspectral Artificial Intelligence: When nonconvex modeling meets hyperspectral remote sensing. IEEE Geoscience and Remote Sensing Magazine, 2021
(3)Meta-Learning Based Hyperspectral Target Detection Using Siamese Network. IEEE Transactions on Geoscience and Remote Sensing, 2022
Reviewer 2 Report
Manuscript deals with hyperspectral imaging algorithms for fake currency detection. The topic is intersting but I am sure there are many other "low cost" solutions. Therefore I would like to ask authors: where is the novelty.
Moreover, authors just mentioned approaches and results are poorly described.
- The methodology should be described in more detailes.
- Conclusion must be proved by results (Fig 6 and Fig 7 are very poor).
The quality of the manuscript corresponds to international conference, not to SENSORS journal.
Round 2
Reviewer 1 Report
No more comments
Reviewer 2 Report
The authors taken into account all my comments, therefore I recommend to accept the manuscript in present form.